# Gold Medals, Silver Medals, Bronze Medals, and Total Medals: An Analysis of Summer Paralympic Games from 1992 to 2016

**DOI:** 10.3390/healthcare10071289

**Published:** 2022-07-12

**Authors:** Miguel Jacinto, Diogo Monteiro, Rui Matos, Raul Antunes

**Affiliations:** 1Faculty of Sport Sciences and Physical Education, University of Coimbra, 3004-531 Coimbra, Portugal; miguel.s.jacinto@ipleiria.pt; 2Life Quality Research Centre (CIEQV), 2411-901 Leiria, Portugal; diogo.monteiro@ipleiria.pt (D.M.); rui.matos@ipleiria.pt (R.M.); 3ESECS—Polytechnic of Leiria, 2411-901 Leiria, Portugal; 4Research Center in Sport Sciences, Health Sciences and Human Development (CIDESD), 5001-801 Vila Real, Portugal; 5Center for Innovative Care and Health Technology (ciTechCare), Polytechnic of Leiria, 2411-901 Leiria, Portugal

**Keywords:** athletes, disability, Paralympic Games, performance indicators, sports

## Abstract

The Paralympic Games (PG) are considered one of the biggest events in the world, with increasing coverage by media and participation. The present study aimed to investigate the variation in the number of gold, silver, bronze, and totals medals in the Summer PG from 1992 to 2016. Data related to the results were extracted from the International Paralympic Committee to an SPSS database. Descriptive statistics and Friedman’s two-way analysis of variance by ranks were used to check the differences across medals in seven editions of the Summer PG, with the correspondent effect sizes. There was a peak in the maximum number of any type of medal between the 1996 and 2000 Summer PG and a decrease until 2008. After that, the number of any kind of medals has been increasing again. There were also significant differences with intermediate to large effect sizes when comparing more distant PG with more recent events. Several external factors can influence performance indicators (e.g., the number of medals) in a negative or positive way. An increase in the number of participants and a greater and better investment by the countries may explain part of our results. The preparation of an athlete must be based on a multidisciplinary team, and future organizing countries must take into account reports of previous events.

## 1. Introduction

The Paralympic Games (PG), considering both the Summer and Winter PG, are a movement that was initiated during the Second World War in rehabilitation centres [1]. In these centres, where soldiers returned with various types of injuries, doctors used sport as a means of rehabilitating the body and mind and of contributing to reintegration into society [2,3,4]. In the space around the rehabilitation centres where small competitions were practiced, they quickly evolved into regional, national, and international events with the format that is now known worldwide as the PG [5,6]. The first PG took place in Rome in 1960 and involved 400 athletes from 23 different countries who competed in 57 medal events across eight sports [7].

On 22 September 1989, the International Paralympic Committee (IPC) was founded, being responsible for directing the Paralympic Movement, with the mission of guaranteeing and supervising its organization [8]. According to the IPC, the main objectives of the PG are: (i) to allow Paralympic athletes to achieve their best performance at the highest level of competition through adequate and adapted conditions and services in a solid environment; (ii) to provide visibility, differentiation, and opportunities that promote and demonstrate the spirit and values of the Paralympic Movement (courage, de-termination, inspiration, and equality); and (iii) to promote social development and leave a positive framework that benefits communities in the host country and around the world [8].

The PG take place every four years, immediately after the respective Olympic Games, in the same city that hosts them [8] and represent an opportunity to convey messages of peace, justice, integration, resilience, values, culture, resistance to the adverse effects of events, inclusion, change in attitudes, stereotyped behaviours, an environment free of urban and architectural barriers, and the greatest achievement for an athlete [4,9], with the number of participants increasing from event to event [10].

The 1992 Barcelona PG represented a turning point, as they presented a new classification concept [11], taking into account their health and functioning (according to the International Classification of Functioning, Disability, and Health) and sport participation, in contrast to a purely medical model [12,13]. This model is based on three steps: (i) determining if an athlete has an eligible impairment; (ii) determining if the athlete meets the minimum impairment criteria for a sport; and (iii) deciding an athlete’s sports class [14]. This allows the athlete to participate on equal terms with other athletes [11]. These athletes go through an evidence-based classification process and are grouped into classes, depending on the degree of limitation and the modality practiced [11,14,15]. Nowadays, according to the IPC [14], the PG involve athletes with a series of deficiencies, namely, impaired muscle power, impaired passive range of movement, limb deficiency, leg length difference, short stature, hypertonia, ataxia, athetosis, vision impairment, and intellectual impairment.

There are currently 22 Paralympic sports held in the Summer PG [16]. Due to the different modalities, there are sports with only one class and others with up to 50 [14], with a high number of events. Subsequently, the first, second, and third place in each class are awarded gold, silver, and bronze medals, respectively.

During the first editions of the Summer PG (between 1960 and 1984), we saw an increase in the number of medals (from 113 to 975), mainly caused by the increase in classes and sporting events. Subsequently, there was a decrease in the number of medals between the 1984 and 1992 Summer PG (from 975 to 489) that was caused by a reduction in the number of sporting events. After this phase of decline, the number of medal events and sporting events remained stable. This number of medals remained stable, despite a slight increase in classes, due to the fact that some sports combined classes in the same event, while others excluded classes from certain events [17].

Previous studies mention that there is a clear advantage at the country level (*p* < 0.05) when it plays at home in the PG [18]. However, the causes are not clear, and further investigation is recommended. More recently, when investigating the Summer PG using a standardized measure of success, some authors confirmed that home advantage is predominant for countries (*p* < 0.01) and in some specific sports (*p* < 0.05) [19].

Knowing that the Paralympic Movement has grown and the number of athletes has increased and that the number of medals is related to the number of sporting events (stable since 1992) and to the number of classes (a slight increase), as far as we know, no study has analysed how the performance has evolved over the events. Therefore, the present study intends to investigate the variation in gold, silver, bronze, and total medals by event in the summer games from 1992 to 2016.

## 2. Materials and Methods

The results of each edition of the Summer PG between 1992 and 2016 were obtained by experienced researchers (M.J., R.A., and D.M.) through the results history file of the IPC [20] and downloaded to the software SPSS. For our analysis, we considered only the countries that were awarded gold in each of the seven selected editions (*n* = 28).

### Statistical Analysis

Descriptive statistics, including the median, mean, standard deviation, and interquartile range, were calculated for all variables. The Shapiro–Wilk (*n* < 50) test was used to verify data normality. In addition, a Friedman's two-way analysis of variance by ranks was used (since the normality was not verified) to check the differences across gold, silver, and bronze medals in seven editions of the Summer PG (i.e., 1992 to 2016), with the correspondent effect sizes for nonparametric tests (Eta squared, η2) [21]. The interpretation of the effect size using η2 was based on the following criteria: <0.01, no effect; 0.01–0.04, small effect: 0.06–0.11, intermediate effect: and 0.14–0.20, large effect [22]. The significance level to reject the null hypothesis was set at 5% [23]. The analyses were conducted using IBM SPSS, version 27.0.

## 3. Results

Table 1 presents the descriptive statistics of the Summer PG (1992–1996), with means, standard deviations, medians, interquartile ranges, and total medals values.

We can verify that the peak of each of the analysed variables occurred in 1996, except for the gold medals, which had their peak in the year 2000 (Table 1 and Figure 1, Figure 2, Figure 3 and Figure 4). After that, there was a decrease until the PG of 2008, as the number of gold and silver medals has been increasing since then, although far from the values previously achieved (Table 1 and Figure 1, Figure 2, Figure 3 and Figure 4).

The figures show some stabilization in the number of medals across the seven analysed editions.

Table 2 presents the differences across gold, silver, and bronze medals in seven editions of Summer PG (i.e., 1992 to 2016) with the corresponding effect sizes for nonparametric tests.

For gold medals, there were differences with intermediate effect sizes, essentially for the 1996 Summer PG compared to the 2004 to 2016 Summer PG. Likewise, there were differences with intermediates effect sizes when comparing the 2000 Summer PG with 2008 and 2016. Knowing that gold medals have a greater “weight”, 1996 was the edition with a greater preponderance compared to the last four editions.

Regarding silver medals, there were differences with intermediate effect sizes when comparing the results of the 1992–2008, 1996–2008, 1996–2016, 2000–2008, and 2004–2008 Summer PG.

Considering bronze medals, there were differences with intermediate effect sizes when comparing the 1992–2008, 1996–2008, and 2000 Summer PG and the last three events.

Likewise, when analysing the total number of medals, differences with intermediate to large effect sizes were verified for the same events as the bronze medals.

## 4. Discussion

The present study aims to investigate the variation in gold, silver, bronze, and total medals in the Summer PG from 1992 to 2016 in a selected sample (countries that were awarded gold in each of the seven selected editions).

Through the performed analyses, there was a peak in the number of any type of medal between the 1996 and 2000 Summer PG and a decrease until the year 2008. Apparently, this number of any type of medal has been increasing again. Likewise, the differences were significant when comparing more distant PG with more recent ones.

Given that the number of events, countries, and participants has increased, a greater distribution of medals may justify the results obtained in this study. However, the literature had already shown that the various indicators (e.g., medals) were stable [16], which is something we confirmed in Figure 1, Figure 2, Figure 3 and Figure 4. If we analyse the editions separately, we see that there was a slight decrease in the number of medals from 2004, which can be interpreted in light of several factors that will be discussed below. Several reports, which are mostly unofficial, reveal the existence of various problems with doping and fraud in sport (classification) at the PG [24]. Since 2004, these issues have been overseen by the IPC [25] with greater rigor and control [26,27]. This fact may have a decreasing effect on the analysed variables (medals won).

Another factor to consider is the prevalence of various injuries and illnesses between the 2000 and 2008 PG [28,29]. The sample of countries selected for the analysis in our study may have been affected by these issues in the period in which there was a decrease in medals.

People with disabilities were rarely seen in the media and in public spaces before the PG. The Paralympic Movement acted as a showcase for its participants, giving visibility and helping to challenge stereotypes and stigmas as well as opening new doors [30]. However, the media can promote both positive and negative/stigmatizing images about people with disabilities [31,32,33,34], and they should be aware of their importance for the Paralympic Movement. Some athletes might not be prepared for such exposure, and their performance may be affected.

This increase in media exposure, greater interest and expectations for sports results (pressure), a high-intensity training environment, and below-average performances lead athletes to exhaustion, physical and psychological injuries, and lower incomes [35,36,37]. As the intensity of training and the number of sporting events increase (qualifications for the PG), life beyond sport is impaired, and this can be another destabilizing factor for sporting performance [38,39,40].

In the same sense, the effects of continuous support for athletes, together with the expectations of institutions/organizations/countries, parents, media scrutiny, and the thought that employment will depend on sports results, can lead to coach exhaustion, which impairs the quality of training and, subsequently, the athletes’ performance [41,42,43,44].

Likewise, several factors may be contributing to the decrease in the number of barriers to the practice of physical activity [45] and the increase in the number of people enrolled in this practice [46], which positively impacts the number of people participating in the PG and performance indicators. Although the interest in the psychological skills of athletes started in the 1980s [47], the development of psychological skills training programs in Paralympic sports is recent [48,49,50]. It is a systematic and consistent practice of mental or psychological skills aimed at improving performance and increasing pleasure and physical activity self-satisfaction and includes training in arousal regulation, imagery, self-talk, goal setting, and concentration [50]. A multidisciplinary team is important before the competition, during the competition, and after the competition, controlling all the issues that can negatively influence an athlete and providing the ideal conditions for the Paralympic athlete to be able to be at the fullest of their performance indicators in the events of the PG. As an example, Israel only started providing psychological support to its athletes in 2000 [51].

Currently, we observe an increase in the social and scientific interest in the athlete’s personality, preparation, and sports results [5,52,53], and the training of psychological skills must be integrated with physical training, enhancing both variables [50,54]. This fact may be one of the reasons why the values of the number of medals won are increasing again.

Another issue that may justify the fact that the value seems to be increasing again is the existence of previously published literature that assesses the risk and identifies critical success factors (public health, surveillance, assessment and control, safety, environmental health, outbreaks of infectious diseases, weather conditions, travel information—public transport, economic assessment, and allergies, among others) that will help to formulate recommendations and organizational plans that manage the identified risks [55,56,57,58,59]. This aspect is essential, not only for the organizing country but also for the entourage that will participate in the PG, since the knowledge of the conditions they will encounter is complete and allows them to prepare accordingly [58,59,60]. A simple example is the fact that weather conditions, namely, temperature and humidity, can influence the performance of athletes [61,62,63]. A prior knowledge of these issues allows adjusting and adapting the structure of the training plan to the conditions that the athlete will find in the competition and the reorganization of future events.

An individual’s eating strategies can significantly influence their physical performance [64]. A nutritional plan for athletes should be differentiated according to their disability, sport, frequency and intensity of exercise, and occupation [65,66,67,68]. Correct nutrition guarantees energy needs and recovery during and after exercise [37], while insufficient nutrition can increase the risk of injury and illness and decrease performance [69]. Despite being scarce, studies on the nutritional knowledge of athletes with disabilities and its relationship with sport have increased [37,70,71,72], as this is also a key element for performance indicators.

There are still some studies that address that the game/modality itself has evolved in technical, tactical, and physical terms over the years, a fact that has been proven in the PG [73,74,75], which is currently considered a high-performance sport. The use of technology in Paralympic sport created tools capable of eliminating disadvantages or measuring them and compensating for performance [76] as well as the development of more, better adapted, and more accurate tools to assess physical [77] and physiological [78] responses and may justify an increase in the number of medals.

Reducing the number of medals may not mean a decrease in performance. The study by Hassani et al. [79] did not find differences in the performance of amputees from 2004 to 2012. Since Paralympic sport has increased interest, increased participation (more athletes and new participating countries) and a greater commitment by countries make the distribution of medals among the countries greater [80]. In this way, the usual winners, namely, the countries analysed in this study, are faced with greater and better competitiveness, and the reduction in the number of medals is not necessarily negative when analysing the PG events.

This document is aligned with the studies on different performance indicators in the PG, where the variation in medals over the last seven editions of the event was studied. Our results point to variations in the number of medals by the sample we selected for analysis. There are several factors (independent variables) that can justify it. However, after the starting point of our study, this conclusion will have to be reached by future studies that study the countries’ individual paths. Likewise, more qualitative studies, focusing on athletes, are needed to understand all the issues around preparation, competition, and the experience after participation in the PG.

In addition, future studies should analyse the moderating effect of the organizing country or continent on the number of participants compared to the number of medals.

## 5. Conclusions

Considering the seven editions of the Summer PG from 1992 to 2016, the present study intends to investigate the variation in gold, silver, bronze, and total medals. There was a variation, with the 1996 and 2000 editions having the highest number of medals won by the sample of this study. A decrease followed. However, it is possible to verify an increase in the last editions.

Faced with high-performance events, the number of participants increasing in the PG, and a greater commitment by countries, we will witness greater competitiveness among athletes, which could lead to a greater distribution of medals, depending on countries, reducing the number of those who can always earn one or more.

Pre- and post-PG reports are also essential to create knowledge that can help in the development of strategies to be adopted during preparations and in future events and to minimize external factors that can negatively affect athletes’ performance.

## Figures and Tables

**Figure 1 healthcare-10-01289-f001:**
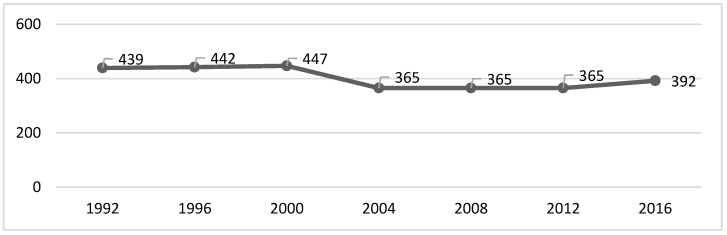
Gold medals evolution all over the editions.

**Figure 2 healthcare-10-01289-f002:**
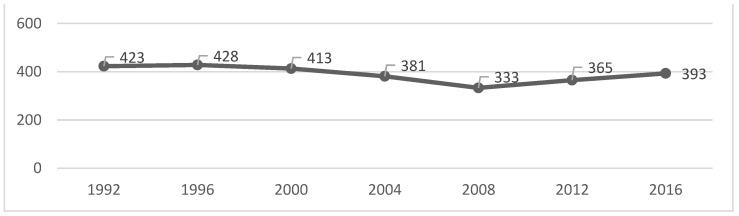
Silver medals evolution over all the editions.

**Figure 3 healthcare-10-01289-f003:**
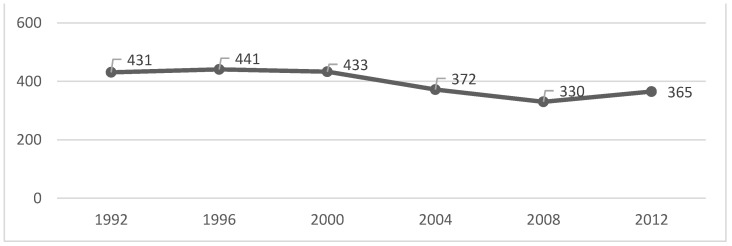
Bronze medals evolution over all the editions.

**Figure 4 healthcare-10-01289-f004:**
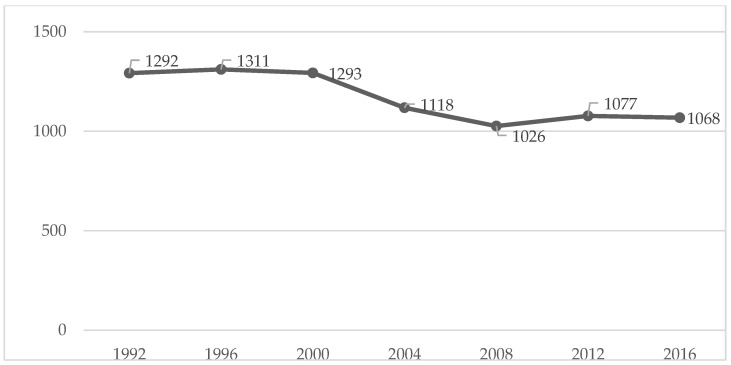
Total medals evolution over all the editions.

**Table 1 healthcare-10-01289-t001:** Descriptive statistics of Summer PG (1992–2016).

Variables	Mean	SD	Median	IQ	S_W	Total Medals
Gold 1992	15.68	3.48	9.0	18	*p* ≤ 0.05	439
Gold 1996	15.79	2.63	9.5	16		442
Gold 2000	15.96	2.97	8.5	22		447
Gold 2004	13.04	2.53	8.0	15		365
Gold 2008	12.96	3.42	5.0	12		365
Gold 2012	13.04	3.49	8.0	9		365
Gold 2016	14.00	4.29	8.0	11		392
Silver 1992	15.11	2.98	9.5	19	*p* ≤ 0.05	423
Silver 1996	15.29	2.85	9.5	16		428
Silver 2000	14.75	2.61	8.0	22		413
Silver 2004	13.61	2.32	9.5	21		381
Silver 2008	11.89	2.85	5.0	16		333
Silver 2012	13.04	2.85	7.5	13		365
Silver 2016	14.04	3.35	7.0	17		393
Bronze 1992	15.39	3.00	10.5	19	*p* ≤ 0.05	431
Bronze 1996	15.75	3.05	10.5	18		441
Bronze 2000	15.46	2.56	10.0	17		433
Bronze 2004	13.29	2.29	7.0	22		372
Bronze 2008	11.79	2.28	6.0	17		330
Bronze 2012	13.04	2.79	7.5	14		365
Bronze 2016	12.93	2.50	7.5	13		362
Total Medals 1992	46.14	9.26	32.5	58	*p* ≤ 0.05	1292
Total Medals 1996	46.82	8.30	30.0	46		1311
Total Medals 2000	46.18	7.81	27.5	65		1293
Total Medals 2004	39.93	6.75	22.5	59		1118
Total Medals 2008	36.64	8.35	16.0	42		1026
Total Medals 2012	38.46	8.99	20.5	31		1077
Total Medals 2016	38.14	9.79	19.5	29		1068

Legend: SD—standard deviation; IQ—interquartile range; S_W—Shapiro-Wilk.

**Table 2 healthcare-10-01289-t002:** Variation in medals for the Summer PG (1992–2016).

Medals	F_r_ Value	Z	*p*	η^2^
**Gold**				
1992–1996	F_r_ = 17.98; *p* ≤ 0.01	−0.91	0.12	-
1992–2000		−0.29	0.62	-
1992–2004		0.34	0.54	-
1992–2008		1.01	0.07	-
1992–2012		0.70	0.23	-
1992–2016		0.09	0.12	-
1996–2000		0.63	0.28	-
1996–2004		1.25	0.03	0.06
1996–2008		1.93	≤0.01	0.13
1996–2012		1.60	0.01	0.09
1996–2016		1.80	≤0.01	0.12
2000–2004		0.63	0.28	-
2000–2008		1.30	0.02	0.06
2000–2012		0.98	0.09	-
2000–2016		1.18	0.04	0.05
2004–2008		0.68	0.24	-
2004–2012		0.36	0.54	-
2004–2016		0.55	0.34	-
2008–2012		−0.32	0.58	-
2008–2016		−0.13	0.83	-
2012–2016		0.20	0.73	-
**Silver**				
1992–1996	F_r_ = 13.66; *p* = 0.03	0.46	0.42	-
1992–2000		−0.09	0.88	-
1992–2004		0.02	0.98	-
1992–2008		1.36	0.02	0.07
1992–2012		0.52	0.37	-
1992–2016		0.66	0.25	-
1996–2000		0.38	0.52	-
1996–2004		0.48	0.40	-
1996–2008		1.82	≤0.01	0.12
1996–2012		0.98	0.09	-
1996–2016		1.13	0.05	0.05
2000–2004		0.11	0.85	-
2000–2008		1.45	0.01	0.08
2000–2012		0.61	0.29	-
2000–2016		0.75	0.19	-
2004–2008		1.34	0.02	0.06
2004–2012		0.50	0.39	-
2004–2016		0.14	0.81	-
2008–2012		−0.84	0.15	-
2008–2016		−0.70	0.23	-
2012–2016		0.14	0.81	-
**Bronze**				
1992–1996	F_r_ = 14.53; *p* = 0.02	−0.16	0.78	-
1992–2000		−0.75	0.19	-
1992–2004		0.23	0.69	-
1992–2008		1.23	0.03	0.05
1992–2012		0.52	0.37	-
1992–2016		0.55	0.34	-
1996–2000		−0.59	0.31	-
1996–2004		0.39	0.50	-
1996–2008		1.40	0.02	0.07
1996–2012		0.68	0.24	-
1996–2016		0.71	0.21	-
2000–2004		0.98	0.09	-
2000–2008		1.98	≤0.01	0.14
2000–2012		1.27	0.03	0.06
2000–2016		1.30	0.02	0.06
2004–2008		1.10	0.08	-
2004–2012		0.29	0.62	-
2004–2016		0.32	0.58	-
2008–2012		−0.71	0.22	-
2008–2016		−0.68	0.24	-
2012–2016		0.04	0.95	-
**Total Medals**				
1992–1996	F_r_ = 19.23; *p* ≤ 0.01	−0.27	0.64	-
1992–2000		−0.66	0.25	-
1992–2004		0.36	0.54	-
1992–2008		1.41	0.02	0.07
1992–2012		0.77	0.18	-
1992–2016		0.77	0.18	-
1996–2000		−0.39	0.50	-
1996–2004		0.63	0.28	-
1996–2008		1.68	≤0.01	0.1
1996–2012		1.03	0.08	-
1996–2016		1.04	0.07	-
2000–2004		1.01	0.08	-
2000–2008		2.07	≤0.01	0.15
2000–2012		1.43	0.01	0.07
2000–2016		1.50	0.01	0.08
2004–2008		1.05	0.06	-
2004–2012		0.39	0.47	-
2004–2016		0.41	0.48	-
2008–2012		−0.64	0.27	-
2008–2016		−0.64	0.26	-
2012–2016		0.01	0.55	-

Legend: F_r_—Friedman test; Z—Z-Score; *p*—*p*-value; η^2^—Eta squared.

## Data Availability

https://www.paralympic.org/results/historical (accessed on 1 June 2022).

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
