# Peer review of "Gold Medals, Silver Medals, Bronze Medals, and Total Medals: An Analysis of Summer Paralympic Games from 1992 to 2016"

_healthcare, 2022, doi:10.3390/healthcare10071289_

Round 1
Reviewer 1 Report
I would like to express my gratitude regarding the opportunity to review this manuscript.
It is an interesting study, congratulations, but at this stage still requiring improvements. Below suggestions with line indication:
5-6 - Please provide authors initials close to the emails and zip codes in affiliations (please check journal guidelines).
10 – Please insert phone contact.
20-24 – The text is related to performance (which is associated to biomechanics, physiology, psychology,…), but the study not. Reformulation is suggestion in the end of the abstract with focus in results and conclusion regarding this specific study.
28 – Please abbreviate PG and afterwards in the text only “PG” (for example 44).
50 – Before 1992, please consider detailing the beginning of PG, number of sports, gender participation, classification, and others. This would provide readers with a clearer and more comprehensive view of the topic under study.
67-74 – “summer” (also in title and other lines of the document). If the authors option is to refer in the text both PG and Summer PG, this could not be clear to readers and leads to the need to describe the winter PGs. Please consider and review.
89 – I believe it is a mistake, “For”.
Please indicate which authors retrieved the data and respective dates. Also experience of the authors.
106 & 116 (tables) – Please provide the results with the same decimals.
Figure or figures are suggested to provide readers with an immediate visual perspective of medals analysis.
127 “considering”? – Please review in this line and the English throughout the manuscript.
134 – Please describe “selected sample”.
Please consider reorganizing the discussion section text aiming readers analysis and interpretation (some paragraphs with few text, contrary to others).
Topics such as the number of sports in each edition, number of participants, gender, classification are more relevant to the discussion compared to nutrition and psychology (more related to performance). Please consider text reformulation.
237-244 – This content seems more related to the discussion section, not conclusions. Please consider analyzing where the conclusion section text best fits (in conclusions currently too much text).
257 – Institutional Review Board Statement, Data Availability Statement and Acknowledgments: are missing.
References format should be carefully reviewed and corrected. They are not according to the journal instructions.
Author Response
I would like to express my gratitude regarding the opportunity to review this manuscript.
It is an interesting study, congratulations, but at this stage still requiring improvements. Below suggestions with line indication:
Response: Thank you very much for the comments and appreciations to the manuscript.
5-6 - Please provide authors initials close to the emails and zip codes in affiliations (please check journal guidelines).
Response: The information has been entered. Thanks for the suggestion.
10 – Please insert phone contact.
Response: The information has been entered. Thanks for the suggestion.
20-24 – The text is related to performance (which is associated to biomechanics, physiology, psychology,…), but the study not. Reformulation is suggestion in the end of the abstract with focus in results and conclusion regarding this specific study.
Response: Thank you for your comment, terminology has been revised throughout the text.
28 – Please abbreviate PG and afterwards in the text only “PG” (for example 44).
Response: Thanks for the comment. The abbreviation was used throughout the document.
50 – Before 1992, please consider detailing the beginning of PG, number of sports, gender participation, classification, and others. This would provide readers with a clearer and more comprehensive view of the topic under study.
Response: Thank you for the comment, a paragraph was inserted in the introduction.
67-74 – “summer” (also in title and other lines of the document). If the authors option is to refer in the text both PG and Summer PG, this could not be clear to readers and leads to the need to describe the winter PGs. Please consider and review.
Response: Na explanation has been posted for when we are referring to PG in general and PG in summer games.
89 – I believe it is a mistake, “For”.
Response: Corrected. Our apologies for the mistake.
Please indicate which authors retrieved the data and respective dates. Also experience of the authors.
Response: It was indicated who collected the data in the respective methodology. Regarding experience, the authors of this work have previous experience in this methodology:
- Rodrigues, F., Mageau, G., Lemelin, E., Vitorino, A., Teixeira, D.S., Cid, L., & Monteiro, D. (2022). Life satisfaction of Paralympians: The role of needs satisfaction and passion. International Journal of Sports Science and Coaching, 17 (3), 510-518. https://doi.org/10.1177/17479541211036224
- Teixeira, D. S., Rodrigues, F., Vitorino, A., Cid, L., Bento, T., Evmenenko, A., Macedo, R., Morales, V., & Monteiro, D. (2021). The Dualistic Model of Passion in Adapted Sport: A double-serial mediation analysis on satisfaction with life. Current Psychology. https://doi.org/10.1007/s12144-021-02186-5
- Cid, L., Vitorino, A., Bento, T., Teixeira, D.S., Rodrigues, F., & Monteiro, D. (2019). The Passion Scale - Portuguese Version: Reliability, Validity and Gender and Sport Invariance. Perceptual and Motor Skills, 126(4), 694-712; https://doi.org/10.1177/0031512519849744
- Scifo, L., Borrego, C., Monteiro, D., Matosic, D., Bianco, A., & Alesi, M. (2019). Sport Intervention Programs (SIP) to Improve Health and Social Inclusion in People with Intellectual Disabilities: Systematic Review. Journal of Functional Morphology and Kinesiology, 4, 57; https://doi.org/10.3390/jfmk4030057
- Vitorino, A., Monteiro, D., Moutão, J., Morgado, S., Bento, T., & Cid, L. (2015). Adapted Physical Activity in the Population with Special Needs. Revista da Federação Portuguesa de Desporto para pessoas com deficiência, 1 (1), 1-5.
- Amorim, A., Travassos, B., Monteiro, D., Baptista, L., Duarte-Mendes, P. (aceite). Efectos de un programa de visulizatión mental en el rendimiento de atletas de Boccia federados y no federados: Efectos de un programa de visulizatión en de Boccia. Cuadernos de Psicologia del Deporte
- Mira, T., Monteiro, D., Costa, A., Matos, R., & Antunes, R.* (2022). Tokyo 2020: A Sociodemographic and Psychosocial Characterization of the Portuguese Paralympic Team. Healthcare, 10, 1185. https://doi.org/10.3390/healthcare10071185
- Matos, R., Monteiro, D., Antunes, R., Mendes, D., Botas, J., Clemente, J., & Amaro, N. (2021). Home-Advantage during Covid-19: An analysis in Portuguese football league.International Journal of Environmental Research and Public Health, 18, 3761. https://doi.org/10.3390/ijerph18073761.
- Jacinto, M., Matos, R., Alves, I., Lemos, C., Monteiro, D., Morouço, P., & Antunes, R.* (2022). Physical activity, exercise, and sport in individuals with skeletal dysplasia: what is known about its benefits?. Sustainability. 14(8), (*Corresponding Author). https://doi.org/10.3390/su14084487.
106 & 116 (tables) – Please provide the results with the same decimals.
Response: The results were insert with the same decimals.
Figure or figures are suggested to provide readers with an immediate visual perspective of medals analysis.
Response: Thank you for your suggested. A figure was inserted to facilitate the visualization of the number of medals (figure 1)
127 “considering”? – Please review in this line and the English throughout the manuscript.
Response: Corrected. Our apologies for the mistake.
134 – Please describe “selected sample”.
Response: Thank you very much for the comment. the information was added to the manuscript (“only the countries awarded that were always stained with gold in the seven selected editions”).
Please consider reorganizing the discussion section text aiming readers analysis and interpretation (some paragraphs with few text, contrary to others).
Response: Thank you for your consideration. The discussion has been revised in order to mention aspects that can negatively and positively and positively influence indicators performance.
Topics such as the number of sports in each edition, number of participants, gender, classification are more relevant to the discussion compared to nutrition and psychology (more related to performance). Please consider text reformulation.
Response: Thanks for the comment, a sentence has been added that addresses these issues.
237-244 – This content seems more related to the discussion section, not conclusions. Please consider analyzing where the conclusion section text best fits (in conclusions currently too much text).
Response: We thank the reviewer for his/her comment. The conclusion section was reformulated according to the reviewer's suggestion.
257 – Institutional Review Board Statement, Data Availability Statement and Acknowledgments: are missing.
Response: These questions do not apply to our study.
References format should be carefully reviewed and corrected. They are not according to the journal instructions.
Response: We appreciate the comment. References have been revised.
Reviewer 2 Report
This article compares the number of medals in the past seven Paralympic Games and gives some suggestions for the training of athletes. The data of this study is slightly thin, and the author gives possible explanations for the results of the data, but the results of the data still lack important academic and/or practical significance.
1, line 96 and line 94,What was the result of Shapiro-Wilk? These data do not follow a normal distribution?
2, line 116, table 2, Friedman´s two-way analysis of variance by ranks needs to report the overall significance first, that is, the main effect. If the main effect is significant, the results of the pairwise comparison can be reported. Also, are the results of this study adjusted for multiple comparisons (eg. Bonferroni)?
3, line 193-200, The author analyzes the reasons that affect the results of the data in the discussion section, but some discussions are too divergent and have little relevance to the findings of this paper. It is recommended to grasp the core, or improve the writing logic.
Author Response
This article compares the number of medals in the past seven Paralympic Games and gives some suggestions for the training of athletes. The data of this study is slightly thin, and the author gives possible explanations for the results of the data, but the results of the data still lack important academic and/or practical significance.
Response: Thank you very much for the comments and appreciations to the manuscript.
1, line 96 and line 94,What was the result of Shapiro-Wilk? These data do not follow a normal distribution?
Response: These data do not follow a normal distribution and that can be verified by the data that we included column in table 2 with the Shapiro-Wilk.
2, line 116, table 2, Friedman´s two-way analysis of variance by ranks needs to report the overall significance first, that is, the main effect. If the main effect is significant, the results of the pairwise comparison can be reported. Also, are the results of this study adjusted for multiple comparisons (eg. Bonferroni)?
Response: We thank the reviewer for his/her comment. The main effect of Friedman’s two-way analysis of variance was included in table 2. Regarding the Pairwise comparisons we already included it in submitted version. Please check Table 2 with Friedman test, value de Z, p and Eta Square for non-parametric tests.
3, line 193-200, The author analyzes the reasons that affect the results of the data in the discussion section, but some discussions are too divergent and have little relevance to the findings of this paper. It is recommended to grasp the core, or improve the writing logic.
Response: The discussion was written and revised. Taking into account that the number of medals was stabilized during the period we analysed, we tried to justify with literature factors that may have contributed negatively and positively to this.

Reviewer 3 Report
interesting work. the literature needs to be corrected - item 71 is incorrectly quotedAuthor Response
Interesting work. the literature needs to be corrected - item 71 is incorrectly quoted
Response: Thank you very much for the comments and appreciations to the manuscript.

Round 2
Reviewer 1 Report
Dear authors,
Thank you for considering my suggestions and incorporating them into the manuscript. Minor revision is proposed considering the below suggestions, with line indication.
96 – The suggestion made was to indicate the experience of the researchers, in this particular case in line 96 - were obtained “by experienced researchers” (MJ, RA and DM) - suggestion.
114 – Table 1 – “1992” and “1996” suggested to present the same format as year 2000 and above.
120 - Figure 1 – Please review the x-axis (“Gold” all years, I believe only year is relevant), The legend related to silver only with one dot, and the figure with two, and fundamentally, the values. For example, total medals in the table are never above 1.500 and, in the figure, always above 2.000. Please carefully review the figure.
From 171 forward – Some references seem to be missing, this is very important to very important to be carefully analyzed. Some examples: 31, 32, 35, 38, 41, 42, 48, 55, 56, 61 (also for example 58 before 57).
273 – All references should be carefully reviewed; they are not according to journal template and instructions for authors. Some examples: “;” should be presented instead of “,” between authors; ref 76 year not in bold; ref 77 journal not in italic.
The article is well written from an English perspective, but a final reading is suggested.
Congratulations for the research and keep up the good work.
Author Response
Response to REVIEWER 1
Dear authors,
Thank you for considering my suggestions and incorporating them into the manuscript. Minor revision is proposed considering the below suggestions, with line indication.
Response: Thank you very much for the comments and appreciations to the manuscript.
96 – The suggestion made was to indicate the experience of the researchers, in this particular case in line 96 - were obtained “by experienced researchers” (MJ, RA and DM) - suggestion.
Response: The information has been entered. Thanks for the suggestion.
114 – Table 1 – “1992” and “1996” suggested to present the same format as year 2000 and above.
Response: Corrected. Thank you for your suggested.
120 - Figure 1 – Please review the x-axis (“Gold” all years, I believe only year is relevant), The legend related to silver only with one dot, and the figure with two, and fundamentally, the values. For example, total medals in the table are never above 1.500 and, in the figure, always above 2.000. Please carefully review the figure.
Response: Dear reviewer, we appreciate your suggestion, which we fully agree. The figure has been changed.
From 171 forward – Some references seem to be missing, this is very important to very important to be carefully analyzed. Some examples: 31, 32, 35, 38, 41, 42, 48, 55, 56, 61 (also for example 58 before 57).
Response: Dear reviewer, we believe that the way we look at the references is in accordance with the journal template and instructions for authors. When we cite [30-33] we are referring to references 30, 31, 32 e 33. I send an extract of the instructions for authors of the journal:
“In the text, reference numbers should be placed in square brackets [ ], and placed before the punctuation; for example [1], [1–3] or [1,3]. For embedded citations in the text with pagination, use both parentheses and brackets to indicate the reference number and page numbers; for example [5] (p. 10). or [6] (pp. 101–105).”
273 – All references should be carefully reviewed; they are not according to journal template and instructions for authors. Some examples: “;” should be presented instead of “,” between authors; ref 76 year not in bold; ref 77 journal not in italic.
Response: Corrected. Thank you for your suggested. As referências 76 e 77 dizem respeito a consultas de site.
The article is well written from an English perspective, but a final reading is suggested.
Congratulations for the research and keep up the good work.
Response: We would like to thank you for the opportunity to submit a revised draft of our manuscript. Your comments and suggestions were of the utmost importance to help clarify and improve our work.
